# The Macrophyte Indices for Rivers to Assess the Ecological Conditions in the Klina River in the Republic of Kosovo

**DOI:** 10.3390/plants11111469

**Published:** 2022-05-30

**Authors:** Pajtim Bytyçi, Albona Shala-Abazi, Ferdije Zhushi-Etemi, Giuseppe Bonifazi, Mimoza Hyseni-Spahiu, Osman Fetoshi, Hazir Çadraku, Fidan Feka, Fadil Millaku

**Affiliations:** 1UBT—Higher Education Institution, Kalabria, Street Rexhep Krasniqi Nr. 56, 10000 Prishtina, Kosovo; pajtim.bytyqi@ubt-uni.net (P.B.); hazir.cadraku@ubt-uni.net (H.Ç.); fidan.feka@ubt-uni.net (F.F.); 2Management of Tourism, Hospitality and Environment, University “Haxhi Zeka” in Peja, 30000 Peja, Kosovo; mimoza.hysenispahiu@unhz.eu; 3Department of Biology, Faculty of Mathematics and Natural Sciences, University of Prishtina, Mother Teresa 5, 10000 Prishtina, Kosovo; ferdijezhushi2010@gmail.com (F.Z.-E.); fadil.millaku@uni-pr.edu (F.M.); 4Department of Chemical Engineering, Materials & Environment, Sapienza University of Rome, Via Eudossiana 18, 00184 Rome, Italy; giuseppe.bonifazi@uniroma1.it; 5Department of Biology, Faculty of Arts and Sciences, Niğde Ömer Halisdemir University, 51240 Niğde, Turkey; osmanfetoshi@hotmail.com

**Keywords:** macrophyte, ecological status, pollution, macrophyte indices

## Abstract

Macrophytes are important elements of aquatic ecosystems that grow in or near water. Their taxonomic composition, species diversity, depth, and density are indicators of environmental health; as such, Macrophytes are used to assess the ecological status of water bodies. Under the aim of assessing the ecological status of the Klina River in Kosovo, a survey was conducted at eight sampling sites along the river course to analyze macrophyte composition, diversity, density, and cover. Three samples were collected at each sampling site from early June to late September. The following macrophyte indices were used to assess the ecological status of the river: Macrophyte Index for Rivers (MIR), River Macrophyte Nutrient Index (RMNI), and River Macrophyte Hydraulic Index (RMHI). Our sampling area included the upper reaches of the river where no organic pollution was detected (oligotrophic), the middle reaches where polluted water from farms is discharged into the river, and the lower reaches characterized by heavy organic pollution from settlements and various industrial activities. There is a positive correlation (*p* < 0.05) between water temperature, turbidity, electrical conductivity (EC), total dissolved solids (TDS), orthophosphates (PO_4_^3−^), ammonia (NH_4_^+^), nitrites (NO_2_^−^), calcium (Ca^2+^), and potassium (K+) with plant density, RMNI, RMHI, EQR-RMNI, EQR-RMHI, and MIR. Sodium (Na+) has stronger positive correlation (*p* < 0.01) with RMNI and RMHI indices and negative correlation with EQR-RMNI and EQR-RMHI. Our results show that ecological status along the river varies from high and good upstream to poor, bad, and moderate running downstream.

## 1. Introduction

Today, freshwater habitats and species are among the most threatened ecosystems in the world, but the importance of freshwater species, ecosystems, and the services they provide to human life and well-being is becoming increasingly understood. Assessment of the ecological status of running waters has traditionally focused on the measurement of physicochemical parameters. Measurement of physico-chemical parameters alone does not present a real and complete picture of the ecological status of the running waters, due to the fact that these measurements only reflect water quality at the moment of sampling [1]. This has led to the use of biological quality parameters for freshwater quality monitoring, a method that would show a real and complete picture of the ecological status of the water ecosystems.

Humans have become the main driving force of changes in the biogeochemical cycle [2]. This phenomenon is manifested in the degradation of ecosystem structure and the alteration of evolutionary processes, such as the water cycle and the cycle of biotic components [3]. Anthropogenic influences, such as agricultural activities, can degrade the quality of surface water and make it unsuitable for drinking, sustainable agricultural use, and sustaining biodiversity [4,5,6].

Anthropogenic activities near river valleys have led to changes in the natural characteristics of watercourses and to ecological degradation of water bodies. The activities carried out in the basins, aimed at increasing the economic use of the river valleys, are important factors in the loss of naturalness. It is also known that hydraulic works such as modifying river courses, cutting of meanders and oxbows, and construction of dams and flow-regulating structures lead to degradation of fluvial ecosystems, including natural river structures [7,8].

The need for accurate information on the relationships between species richness and the environment, and between species composition and the environment, has continued to grow in the face of global change [9,10,11].

The introduction of the Water Framework Directive has stimulated the intensive development and improvement of a variety of bioassessment methods in the EU over the last decade [12,13,14,15,16]. For biological monitoring, it is crucial to develop a method that supports comprehensive, quick, and cost-effective surveys and that provides highly accurate data for reliable and unambiguous assessments of the ecological status of water bodies [16,17].

The usefulness of living organisms in detecting environmental change has been frequently confirmed in both terrestrial and aquatic ecosystems [16]. The quality of a habitat can be reflected both in the abundance of individual species and in the structure and diversity of communities [18,19,20].

Aquatic plants are critical for maintaining water transparency by taking up nutrients from the water column and sediment to compete with phytoplankton for nutrients and light [21], by releasing allelopathic substances that can inhibit phytoplankton growth [22], and by promoting sediment stability and reducing resuspension [23,24].

Macrophytes are fundamental to the structure and functioning of river habitats [25], being involved in energy flow, nutrient cycling, and sedimentation processes [26]. 

The presence and diversity of macrophytes in a water body depend on water depth and quality, as well as on water movement and substrate characteristics [27]. They can improve water quality [28] by absorbing nutrients from the water and sediment, thus contributing to the self-purification capacity of the ecosystem [29]. Abundant macrophyte communities can have a significant impact on the daily dynamics of dissolved oxygen and other water quality parameters of a water body [30,31,32,33]. 

The diversity and quantity of macrophytes present in a river can tell us how well that river is functioning. This method is based on the principle that different macrophytes are associated with different amounts of nutrients (especially phosphorus) and flow conditions. Depending on the type of river and its fertility (WDF), different combinations, amounts, and numbers of macrophytes can be expected. So far, the main groups of aquatic organisms, i.e., fish, macroinvertebrates, phytoplankton, and macrophytes, have been used for the assessment. In order to obtain comprehensive information on ecosystem degradation, biological research is supplemented by hydromorphological elements of water bodies and the results of the physicochemical assessment of water samples [8]. 

In order to monitor and assess rivers, many scientific studies have recently been carried out in different parts of the world for biological monitoring and aquatic ecosystem study, based on macrophytes as one of the biological elements for the assessment of the ecological status of rivers required by the EU Water Framework Directive (WFD). In Kosovo, the first studies on water quality assessment based on macrophytes are from the Lepenci River basin [16]. However, many parts of the river basins in Kosovo have not yet been explored on the basis of this component. Since there were no data on the use of macrophytes and their use as indicators of water quality in the Klina River, this study is one of the first aimed at assessing water quality based macrophyte indices based on WFD requirements.

The main pollution factors of the Klina River are considered to be the discharge of sewage without prior treatment, the disposal of various solid wastes on the riverbanks, nutrient-rich agricultural runoff, illegal river irrigation systems, and the drying up of tributaries in summer, as well as the alteration of the riverbed (concrete and large stone slabs).

The main hypothesis of this study was that the water of the Klina River is exposed to a number of pollutants of different natures that affect its quality by altering the physicochemical, hydromorphological, and biological parameters of the water.

The objectives of this study were:○To determine how aquatic nutrients affect macrophyte species diversity, cover, and density in the Klina River.○To use the macrophyte-based indices to classify water quality and measure the ecological status of the river.○To determine water quality based on physico-chemical parameters according to the GD161 standard.

## 2. Materials and Methods

### 2.1. Study Area

The Klina River is one of the tributaries of the largest water basin in Kosovo, the Drini i Bardhë, located in the western part of Kosovo. In terms of hydrography, Kosovo is divided into four river basins: Drini i Bardhë, Ibri, Morava e Binçes, and Lepenci. Kosovo’s rivers flow into three sea basins: the Black Sea, the Adriatic Sea, and the Aegean Sea. The river Klina rises in the village of Kuqicë and crosses the territory in the north-south axis and passes through the entire territory, as well as through the city center of Skenderaj, with a slow flow and medium depth (Figure 1). In the territory of the municipality there is a large number of streams which completely flow into the river [34]. The river continues to flow through the center Klina, and after a few kilometers, it flows into the Drini i Bardhë on the west side. In this part, the river Klina has its greatest width, while in the vicinity of the estuary in the river Drini i Bardhë, it reaches 5 m. With a length of 75 km, the Klina river is the second-biggest tributary of the Drini i Bardhë river basin (Table 1) [34].

The average annual inflow of the river Klina is 2.8 m^3^/s, but from the detailed data recorded by the Hydrometeorological Institute of Kosovo (1961–1984), the average inflow is 1.50 m^3^/s or about 47.3 × 106 m^3^, which, after the river Mirusha, has the smallest inflow among all the tributaries of the river Drini i Bardhë [34].

In its upper part, the river is a typical mountain river with high flow velocity, while near the city of Skenderaj, its flow begins to slow down and spreads out in the alluvial plain with humus (sandy) soil. Throughout its course, the Klina River has almost the character of a narrow canyon [34].

The geological composition of the bed along the middle course and before it flows into the Drini i Bardhe River is alluvial humus.

The river Klina exhibits a Mediterranean regime, with a maximum in March and November and a minimum in May and September. If we compare the minimum inflows with the maximum inflows for any return period, it becomes clear that this river has all the characteristics of a stream (Figure 1). Despite the fact that the catchment has good vegetation cover, the amount of alluvium, whether suspended or dragged, gives the river the character mentioned above. The surface of the Klina river basin up to its middle profile, in the city of Skenderaj, has an area of 77.75 km^2^ [34].

In terms of vegetation cover, the most represented are shrub forests with 60.20%, followed by forests with 14.15%, while meadows, pastures, and agricultural areas are represented with 25.65%. Regarding the erosion processes, 84.45% of the area is listed in category III [34].

### 2.2. Macrophyte Sampling

Macrophyte surveys were carried out using the European national monitoring methodology based on WFD methods. The survey was conducted using standard methods CEN 14184:2003 Water Quality Guidance standard for the surveying of the aquatic macrophytes in running waters (Comitè Europèenne de Normalisation, 2003) [35,36].

The macrophyte samples were taken from eight sampling points at a distance of 100 m along the length of the river, and were all submerged, free-floating, amphibious, and emergent monocotyledonous and dicotyledonous plants, liverworts, mosses, and pteridophytes, as identified. The assessment also included macrophytes growing or rooting on portions of the riverbank that are likely to be inundated for more than 85% of the year. Three samples were collected at each station, from the beginning of June to the end of September, as the vegetation phases of the different species vary (Table 1). Some plant species were immediately identified in the field, while others were identified in the laboratory. Specimens were identified according to the following literature: Flora of Albania [37,38], Flora of Bulgaria [39,40,41,42,43,44], to Flora of Serbia [45,46,47,48], while the nomenclature used accords with the database on the Plant List (theplantlist.org) [49]. In addition, the nomenclature for the scientific plant names, according to The Plant List (working list of all plant species), Euro+Med Plant Base [50], and Flora Europea [51], was used.

The macrophytes were identified to the species level. The presence of each taxon was recorded with its percentage cover using the following nine-point scale according to Holmes et al. [52]: 1 for 0.1%, 2 for 0.1–1%, 3 for 1–2.5%, 4 for 2.5–5%, 5 for 5–10%, 6 for 10–25%, 7 for 25–50%, 8 for 50–75%, and 9 for 75%.

Kohler’s 5-point scale is used to record the species abundance in the sampling points (Kohler,1978): 1 = very rare, 2 = rare, 3 = common, 4 = frequent, 5 = abundant, predominant.

Based on the macrophyte data, five macrophyte metrics were calculated, namely the MIR [29,53,54], RMNI, RMHI, EQR RMNI, and EQR RMHI [16,55,56], from the WFD for evaluation of the ecological status of the rivers.

Macrophyte Index for Rivers (MIR) (Table 2) is calculated with Equation (1):(1)$$\mathrm{MIR}=\frac{\sum_{1-1}^{n}\left(Li*Wi*Pi\right)}{\sum_{i-1}^{n}\left(Wi*Pi\right)}$$
where MIR—value of the Macrophyte Index for Rivers at the sampling site, *n*—number of species at the sampling site, *Li*—indicator value for the *i*-th taxon, *Wi*—weighting factor for the *i*-th taxon, *Pi*—ratio of coverage for the *i*-th taxon [53,57,58].

### 2.3. River Macrophytes Nutrient Index (RMNI)

This index measures the aquatic plants (macrophytes) growth in the river in relation to nutrients. Depending on the amount of nutrients in the water, its value ranges from 1–10 [16,56]. RMNI is calculated with the formula:(2)$$\mathrm{RMNI}=\frac{\sum_{j=1}^{n}\left(CjxRj\right)}{\sum_{J=1}^{n}Cj}$$
where “*Rj*” is the river macrophyte nutrient index score in Column for taxon “*j*”. Here, “*j*” represents a taxon listed in Column, present in the sample and with a value listed in Column, with a value of 1 to “*n*” indicating which of all taxa (total number= “*n*”) listed in Column and present in the sample it represents; “*Cj*” is the taxon cover value for taxon “*j*” determined in accordance with.

### 2.4. River Macrophyte Hydraulic Index (RMHI)

This index measures the macrophytes grow related to the river flow rate and is expressed on a scale from 1–10; the higher the flow rate, the higher the energy and vice versa [16,56]. RMHI is calculated according to the formula:(3)$$\mathrm{RMHI}=\frac{\sum_{j=1}^{n}\left(CjxHj\right)}{\sum_{j=1}^{n}Cj}$$
where “*Hj*” is the river macrophyte hydraulic index score in Column for taxon “*j*”; “*j*” represents a taxon listed in Column, present in the sample and with a value listed in Column, where “*j*” has a value of 1 to “*n*” indicating which of all taxa (total number = “*n*”) listed in Column and present in the sample it represents; “*Cj*” is the taxon cover value for taxon “*j*” identified and.

### 2.5. How Do We Decide the Biological Status?

Ecological Quality Ratio (EQR), is expressed as a numeric value that ranges from 1 (natural or near natural state) to 0 (highly degraded by pollution or other disturbance) [16,56].

This is subdivided equally into the five quality categories, as required by the WFD (Table 3).

The EQR for the parameter RMNI should be calculated using the following Equation:EQRRMNI = (observed value of RMNI − worst possible RMNI) ÷ (reference value for RMNI − worst possible RMNI)(4)

EQR for the parameter RMHI, should be calculated using the following equation:EQRRMHI = (observed value of RMHI − worst possible RMHI) ÷ (Reference value of RMHI − worst possible RMHI)(5)

In parallel with macrophytes sampling, the water samples were taken for physical and chemical analyses.

The parameters such as water temperature (WTemp), air temperature, turbidity (NTU), electrical conductivity (EC), pH, total dissolved solids (TDS), total suspended solids (TSS), dissolved oxygen (DO), and oxygen saturation (OS) were measured on-site using a portable instrument, while other parameters such as biochemical oxygen demand (BOD), chemical oxygen demand (COD), total organic carbon (TOC), nitrates (NO_3_^−^), orthophosphates (PO_4_^3−^), total phosphorous (TP), ammonia (NH_4_^+^), nitrites (NO_2_^−^), sulphates (SO_4_^2^^−^), calcium (Ca^2+^), magnesium (Mg^2+^), sodium (Na^+^), potassium (K^+^), and chlorides (Cl^−^) were analyzed at the Kosovo Institute of Hydrometeorology with standard methods ISO 2014 [60].

### 2.6. Statistical Analysis

We used SPSS 24 to calculate Pearson’s correlation between the macrophyte metrics and environmental variables. The statistical significance of the relationship between macrophyte data and environmental parameters was evaluated using canonical correspondence analysis. CCA was performed using XLSTAT 2018.1.by Addinsoft.

## 3. Results and Discussion

### 3.1. Water Quality Parameters and Ecological Status

Population growth on both sides of the Klina River flow from SP2 to SP8, industrialization, sewage, and industrial wastewater discharged into the river are factors that significantly affect its pollution. To understand and evaluate the water quality of Klina River, the following physico-chemical parameters have been researched in eight localities: water temperature, turbidity, electrical conductivity, total dissolved solids, total suspended solids, dissolved oxygen, dissolved oxygen saturation, biochemical oxygen demand, chemical oxygen demand, total organic carbon, nitrates, (n, n)-diethyltryptamine, orthophosphates, total phosphorus, ammonia, nitrites, sulphate, calcium, magnesium, sodium, potassium, and chloride (Table 4). For analysis and comparison of the results of physico-chemical parameters, the standard for assessment of the ecological status of surface waters from ANEX 1, according to the WFD of the Romanian mandate of 2006 (GD 161), was used. The results revealed that between the sampling stations there were differences in some physico-chemical parameters; most of the parameters were at the average level and showed good river water quality in SP1 and SP2, while in SP3, SP4, SP5, SP6, SP7, and SP8, the ecological condition and water quality was very bad for the parameters measuring nitrates and phosphates—a result of organic pollution from inhabited areas.

The range of WT in our samples was from 12.1 °C in SP1 to 19.1 °C in SP4. The average value with standard deviation obtained for the eight stations was 16.2 ± 2.6 °C. Similar studies have shown that increased temperature affects the growth of macrophytes [61]. Based on our results, we noticed that in the stations where we had higher plant density and higher coverage, higher temperatures were registered; therefore, we can confirm that the increase of temperature affects plant density and the increase of higher coverage.

The higher level of pH can affect the aquatic life at a certain level; however, an optimum level of 7–8.5 was recommended by Bis [62]. The variation of pH ranged from 7.08 in SP6 to 8.26 in SP4. The mean value with standard deviation for the eight stations for pH was 7.27 ± 0.37.

Turbidity (TUR) value showed variation from 2.8 mg∙dm^−3^ in SP1 up to 30.6 mg∙dm^−3^ in SP4. The average value with standard deviation for the eight stations is 20.3 ± 10.3 mg∙dm^−3^. According to other research [63,64,65], the light transmission decreases with increasing turbidity, which leads to changes in community structure and reduction in vegetation diversity. According to Nurminen [66], in clay-turbid eutrophicated lakes, emergent vegetation may play an important role in seasonal and diurnal regulation of zooplankton by providing refuge, especially for free-swimming cladocerans. In addition, according to other studies [67,68], P. crispus affected the richness of epiphytic algae by reducing nutrient concentrations (reduction in total organic carbon, total nitrogen, and chemical oxygen demand) and enhancing water transparency (reduction in turbidity and total suspend solids) to enhance the richness of epiphytic algae.

For electrical conductivity (EC) the lowest value, 376 µS∙cm^−1^, was recorded in SP8, whereas the highest was in SP2, 856 µS∙cm^−1^. The average value with standard deviation for the eight stations was 599.7 ± 179.0 µS∙cm^−1^. According to Rameshkumar et al. [68], electrical conductivity, total dissolved solids, and turbidity have negative impact in Macrophytes.

Total dissolved solids (TDS) showed variation from 184 mg/L in SP8 up to 428 mg/L in SP2. The average value with standard deviation for the eight stations for TDS was 342.3 ± 88.8 mg/L, which is influenced mainly by urbanization, fertilization runoff (agricultural), and domestic effluents.

The total suspended solids (TSS) value showed variation from 1.1 mg/L in SP8 up to 85 mg/L in SP3. The average value with standard deviation for the eight stations is 27.5 ± 28.7 mg/L.

Dissolved oxygen (DO) ranged from 0.3 mg/L in SP3 up to 10.90 mg/L in SP8. The average value with the standard deviation for the three seasons for DO has been 5.22 ± 3.13 mg/L. According to the average values of the eight monitoring stations, comparing them with the standard values (GD161), it turns out that the water in the Klina River belongs to the third class (good quality). The low DO level indicates the degree of pollution in the water bodies [69].

Dissolved oxygen (DO) is an important water quality parameter for sustaining aquatic life, and many organisms are sensitive to low oxygen concentrations in water. Oxygen enters water by diffusion from the air and as a photosynthetic by-product of aquatic plants [68,70].

Biochemical oxygen demand (BOD5) is a measure of the oxygen consumption by microorganisms in the oxidation process of organic matter [71]. Other studies [72] have shown that aquatic plants reduce BOD and COD levels by increasing water quality; also according to this research, macrophytes are capable of removing pollutants from polluted waters, especially N, P, and K. The minimum value of BOD in Klina River in our eight sampling points was 1.5 in SP1, while the maximum value was 42.2 in SP3. The average value with standard deviation for eight stations was 13.5 ± 14.7. Compared to the values of standard GD161, the water in the river belongs to the third category and is of good quality, but it should be noted that in the stations SP2, SP3, SP4, and SP5, with agricultural pollution and sewage discharge, the values of BOD were very high and the water belongs to the Bad category, according to the GD161 standard.

The variation of COD ranged from 4.2 in SP1 to 87 in SP3. The average value with standard deviation for the eight stations for COD was 38.2 ± 30.4. According to the average values of the eight monitoring stations, comparing them with the standard values (GD161), it turns out that the water in the Klina River, according to COD, belongs to the third class and is of good quality. According to certain research [71], water hyacinth was able to decrease COD from its initial value to the final value, below National Environmental Quality Standards (NEQS). Moreover, their result confirmed growth of macrophytes, showing high performance in removing COD, mainly because of well-developed root systems.

Excess of nutrients N and P in the water body caused overproduction of phytoplankton and resulted in O2 depletion [71]. According to research [73], the plants Pontederia parviflora and Typha domingensis can reduce the level of COD.

High levels of COD reduction in phytotreatment systems are quite common and are related to the increase in oxygenation that plants benefit from in the environment and allow greater assimilation of C by microorganisms [74], which was also observed in our studies at site SP6, where the species Typha latifolia G.Mey had a wide distribution.

The variation of total organic carbon ranged from 1.1 in SP1 to 28.8 in SP3. The average value with standard deviation for the eight stations is presented in Table 4.

Total organic carbon can serve as food (carbohydrates, amino acids) or information source [75] for aquatic organisms but has no known effects on freshwater macrophytes [76].

The variation of nitrates ranged from 0.7 in SP3 to 26.2 in SP4. Based on these parameter values, the river Klina is of the third class and of good quality. It has been proven [77] that Macrophytes affect the reduction of nitrates in water, depending on their ability to produce carbon for denitrification [78].

The variation of orthophosphates ranged from 0.011 in SP1 to 0.320 in SP3. Orthophosphate levels place the Klina River in the fifth class, of poor water quality. The values of total phosphorous ranged from 0.046 in SP1 to 1.28 in SP3; the water in the river Klina according to this parameter belong to the second class and is of good quality.

According to Daldorph [79], an increase in phosphates resulted in a significant increase in the biomass of the floating plant Lemna minor and the rootless plant Ceratophylum demersum. In our study, these two plant species were also present in SP2, SP3, SP5, SP6, and SP7, with higher phosphate levels and high biomass. The variation of ammonia ranged from 0.033 in SP1 to 1.650 in SP3, which showed that the water in Klina River based on this parameter belongs to the good quality of the second class.

The nitrites (NO2^−^) in the water varied from 0.021 in SP1 to 3.75 in SP7. According to the standard (GD161), this water is of high quality, class I. The research of Wang et al. (2021), concluded that nitrates have a negative impact on submerged aquatic plants, for example in Myriophyllum spicatum L. Similarly, it was shown in our study in the locality SP6 where Myriophyllum spicatum L. was present, it had no large biomass due to the high value of ammonia—0.847 mg/L.

Many studies have shown toxic levels of ammonia for many plant species [80,81,82,83,84,85].

### 3.2. Richness, Diversity and Cover of Macrophytes

The results of macrophyte composition in the sampling points are presented in (Table 5).

Based on the results analyzed during our research in the Klina River, we identified 67 species of macrophytes belonging to 26 families. The richest station in species was SP1 with 26 species, followed by SP5 with 21 species, SP6 with 19 species, SP4 with 17, SP7 and SP8 with 11 species each, and SP2 with 10 species; the poorest locality was the locality SP3, with 8 species (Table 5).

Station SP1 is a mountainous source area; the localities SP2, SP6, SP7, and SP8 are urban and inter-urban areas, while SP3, SP4, and SP5 are agricultural areas.

Out of the 67 macrophyte species present in the Klina River, the following species were present only in SP1: *Adoxa moschatellina* L., *Amblystegium riparium* (Hedw.) Schimp, *Cardamine amara* L., *Ceratophyllum demersum* L., *Cinclidotus aquaticus* (Hedw.) B. & S., *Cinclidotus fontinaloides* Palisot de Beauvois, *Galium palustre* L., *Lysimachia nummularia* L., *Lythrum salicaria* L., *Marchantia polymorpha L. var aquatic, Mentha longifolia* (L.) Huds., *Nasturtium microphyllum* (Boenn. ex Rchb.) Rchb., *Persicaria amphibia* (L.) Delarbre, *Petasites japonicus* subsp. *Japonicus*, and *Plantago media* L.

The species present only in SP2 was *Persicaria glabra* (Willd.) M.Gómez. The species *Juncus inflexus* L. and *Rumex palustris* Sm were recorded only in SP3.

In SP4, two species were recorded that were not recorded in other sampling points: *Cinclidotus aquaticus* Bruch & W.P.Schimper and *Cratoneuron filicinum* Spruce.

The species recorded only in SP5 were *Callitriche cophocarpa* Sendtn, *Ceratophyllum demersum* L., *Leersia oryzoides* Michx., *Rumex longifolius* DC., and *Scrophularia nodosa* L.

The species *Callitriche stagnalis* Scop., *Helosciadium repens* (Jacq.) Koch., *Myriophyllum spicatum* L., *Potamogeton fluitans* Sm., and *Potamogeton gramineus* L. were recorded only in SP6, whereas Agrostis stolonifera L., Potamogeton nodosus Poir., Ranunculus aquatilis (Dumort.) Ba., *Stratiotes aloides* L., and *Stuckenia pectinata* (L.) Börner were found only in SP7 (Table 5).

The most frequent species were *Lemna minor* (6 localities), *Typha angustifolia* (5) and *Mentha aquatica* (5) (see Table 5).

Data analysis showed that submerged plant density, plant density, and cover had lower values in SP1 (1) and SP8 (1), where the river flow velocity was higher compared to other SPs (Table 6).

Moreover, at SP1 and SP8 pH (7.93–8.11), turbidity (12.1–12.7), and water temperature were at lower values compared to other SPs (Table 4 and Table 5). As a result of industrial pollution, the number of macrophytes, the density of submerged plants, plant density, and cover were low in SP3, where the amount of dissolved oxygen was only 0.03 mg/L and dissolved oxygen saturation was 0.3, while biochemical oxygen demand was of much higher values compared to other SPs.

A cluster analysis with Ward’s method revealed two distinct groups of analyzed elements and values of the indices in the river Klina. The first group consisted of stations SP1 and SP7, with SP1 located in the headwaters and SP7 at the confluence of the Klina River with the Drini i Bardhë River, with high water level and flow velocity, which had a positive effect on water quality (Figure 2).

The second, largest group was constituted of localities SP2, SP3, SP4, SP5, SP6, and SP8, the ones which had a higher density of Macrophyte plants as a result of high organic pollution from settlements and agricultural activities, as well as from industry (Figure 2).

### 3.3. Macrophytes Indices and Ecological Status

According to the values of MIR index, the water quality in stations SP1, SP5, and SP6 is very good and belongs to the first class (I); stations SP2 and SP4 belong to class II of quality (Good), while at stations SP3, SP7, and SP8, the water quality is poor and belongs to the fourth class (IV).

According to the RMNI index (River Macrophyte Nutrient Index), the station richest with nutrients is SP3, with a value of 8.56 where only 8 Macrophyte species were present. It is worth noting that even based on the nitrates, orthophosphates, total phosphorus and nitrites nutrient parameters, according to GD161 standards, water belongs to the fifth category of quality. According to the average values for ammonia, the water belongs to the third category, which is due to various organic pollutants originating from the discharge of domestic waste water, animal manure, and fertilizer from agricultural areas as well as from industrial discharges of some factories which are located around this monitoring station. Higher values of this index are also found in stations SP7, SP2, and SP8, which are urban and interurban areas. The lowest value of this index is found in the source area SP1. The dominant species in these localities that had the highest river macrophyta nutrient index score were *Typha angustifolia, Rumex palustris, Polygonum mite, Sparaganium erectum, Lemna minor,* and *Potamogeton nodosus* etc., which are species that live in places rich in nutrients.

As far as RMHI (River Macrophyte Hydraulic Index) is concerned, the highest values of this index were found at stations SP7 (8.5) and SP3 (8.42), which is due to the high pollution level and slow water flow caused by riverbed alteration and other anthropogenic activities. The Macrophyte density in these monitoring stations is very high, so many species that are calculated with this index are related to low energy velocity, which depends on the velocity of water flow. The lowest value of this index was found at station SP1 (3.57), in the source area, without or with minimum anthropogenic influence, with high water flow and low amount of nutrients.

Based on the RMNI’s Ecological Condition Classification (EQR), Station SP1 has a score of 1, which places the water quality in the first class (I) with high ecological condition. Due to the low anthropogenic influence, this station is taken as the reference site for the calculation of ecological status in other monitoring stations, according to the instructions of the WFD. Stations SP3, with EQR = 0.21, and SP7, with EQR = 0.38, are classified as poor and belong to the fourth class of quality (IV) and poor ecological status. Stations SP2, SP5, SP6, and SP8 belong to the second class of water quality (II) with good ecological status based on their EQR values.

According to the RMHI’s Ecological Condition Classification (EQR) values, station SP1 has a value of 1, meaning that the water quality is in the first class of (I) and has a high ecological condition. Based on the values obtained, stations SP3 and SP7 are classified in the fourth class (IV) of poor water quality and have poor ecological status. In the station SP2 (Llaushë), EQR is 0.57, the water is of the third class of quality (III) and moderate ecological status, while stations SP5, SP6, and SP8, have shown good water quality and belong to the second (II) class, with good ecological status.

Compared to the results in the Lepenci catchment [16], the Klina River has a greater number of macrophyte species and density, which is due not only to the higher organic pollution, but also to the different hydromorphological parameters (composition of the riverbed, flow velocity, river depth, etc.) of these two rivers. Pearson correlation analyses between Macrophyta indices, ecological quality ratio, and plant density with physicochemical parameters show that pH, DO, BOD, COD, TOC, nitrates and MTS have no significant correlation with any of the variables RMNI, RMHI, MIR, EQR-RMNI, EQR-RMHI, or plant density (Table 7).

However, we can see a significant correlation of these variables in the table with water temperature, turbidity, electrical conductivity, total dissolved solids, orthophosphates, ammonia, nitrites, calcium, sodium, and potassium. There is a strong positive correlation (*p* < 0.01) of sodium (Na+) with RMNI, RMHI, and EQR RMNI and EQRRMHI (*p* < 0.05), which means that if sodium were to increase, these parameters will increase as well. A positive correlation (*p* < 0.05) emerged between plant density and water temperature, NTU, TDS, and K (see Table 7).

Our results show that sodium (Na+) positively impacts the nutrient concentration in the water; however, it has negative effect in ecological status based on RMNI and RMHI.

CCA revealed that all tested physicochemical parameters significantly affected the variability of species presence, plant density, and index values (*p* ≤ 0.001). The nutrient parameters (orthophosphates, ammonia, sodium, nitrites) and the physical parameters (water temperature, turbidity, electrical conductivity, and dissolved solids) were found to be the most important (see Figure 3).

## 4. Conclusions and Recommendations

Our results show that the Klina River is exposed to a large number of organic pollutants, which is reflected in the high concentration of nutrients, high presence of macrophytes species, and density throughout the river, which directly affected the values of used Macrophyte indices.

The number and species of macrophytes, as well as the values of the indices used (MIR, RMNI, RMNI (EQR), RMHI, RMHI (EQR)), showed that the water quality in SP1, SP5, and SP6 was very good; stations SP2, SP4, and SP8 had good water quality, while at stations SP3 and SP7, the water quality was poor.

The small number of macrophytes (8 species) as well as the values of indices (RMHI–8.42, EQR–0.21, DO–0.3 mg/L, BOD–42.2, COD–87) show that the water quality in SP3 is very bad and worrying for biodiversity. Therefore, we recommend that local environmental institutions and those at the central level take urgent measures to prevent water pollution of the Klina River, especially in localities SP3 and SP7.

Our research shows that macrophytic indices are good indicators for biological assessment of water quality and are therefore recommended for use in river management plans for biological monitoring in this country. As an urgent measure, we propose the installation of water treatment plants in the Klina River and obliging industrial operators to treat industrial and household wastewater before it is discharged into the river.

## Figures and Tables

**Figure 1 plants-11-01469-f001:**
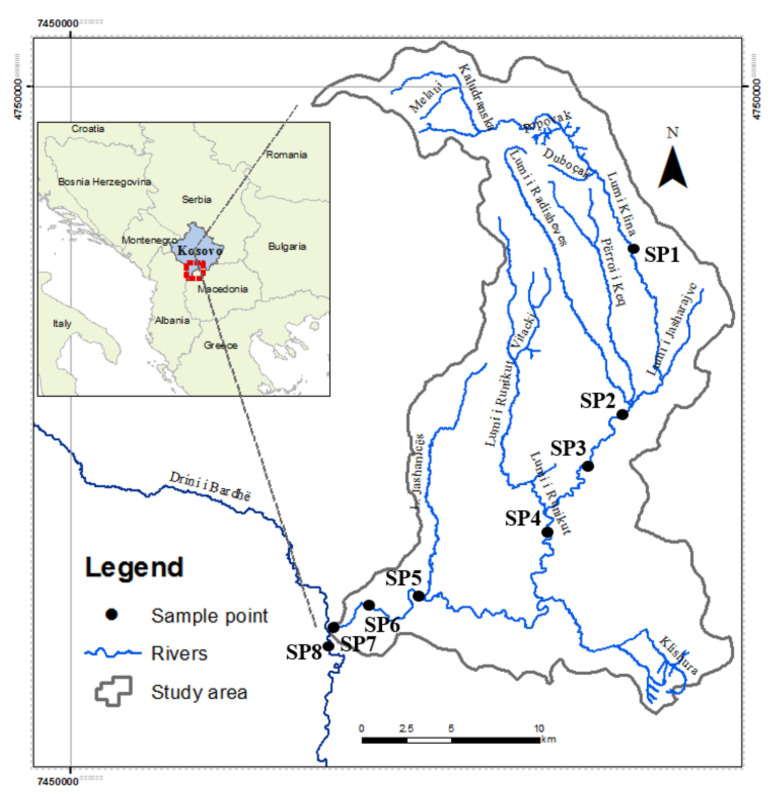
Location of investigated area.

**Figure 2 plants-11-01469-f002:**
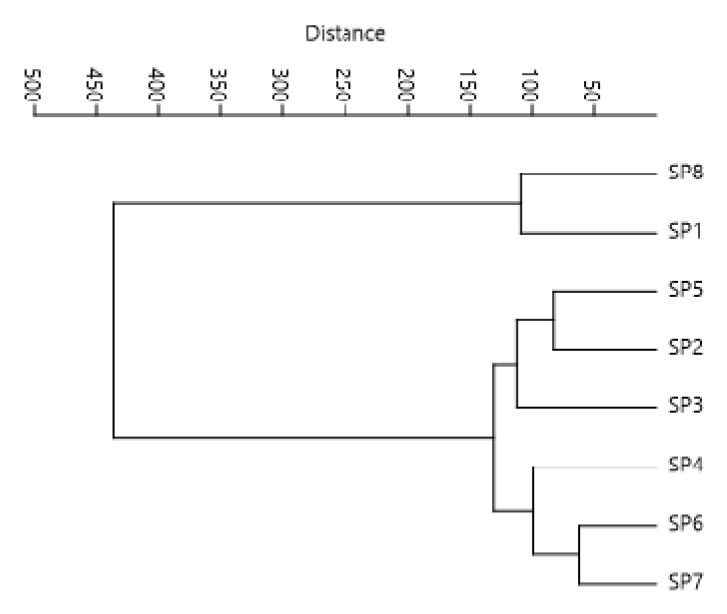
Hierarchical cluster dendrogram for elements and value indices at eight monitoring stations.

**Figure 3 plants-11-01469-f003:**
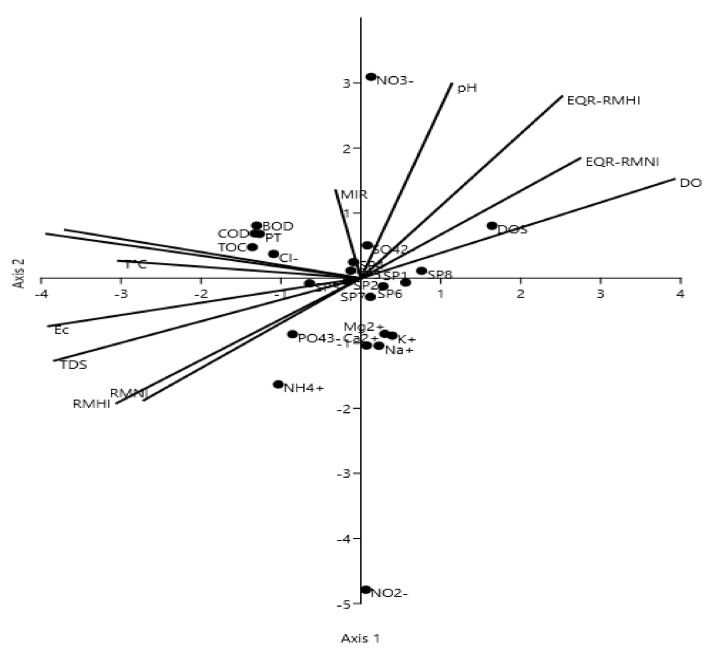
CCA (canonical correspondence analysis) plot showing the relationships between plant density, value indices, and physicochemical parameters.

**Table 1 plants-11-01469-t001:** The sampling sites with geographic coordinates, hydro morphology and Riparian vegetation.

	Sampling Stations	Latitude (N) Longitude(E)	Hydro Morphology	Riparian Vegetation
SP1	The mountain spring of Klina river in village Kuçicë.	42°37′8″ N 20°53′49″ E 1200 m	Natural river bed. Flow velocity 2.5 m/s.	Well-developed on both sides of the riverbank, dominated by deciduous trees Salicaceae, Betulaceae, and Fagaceae.
SP2	Llaushë	42°43′21″ N 20°45′60″ E	Located 1 km from the city of Skenderaj, where the river receives discharges from municipal wastewaters, agriculture and industry. Water flow l.5 m/s, water color is dark with bad odor.	Well-developed on both sides of the riverbank, dominated by deciduous trees Salicaceae.
SP3	Tushilë	44°42′43″ N 20°45′39″ E	The river receives discharges from agriculture and industry. Water flow l m/s, water with bad odor.	Well-developed on both sides of the riverbank, dominated by deciduous trees Betulaceae.
SP4	Açarevë	42°39′6″ N 20°43′15″ E	Riverbed with waste and agricultural runoff due to the land use. Flow velocity 2 m/s.	Deciduous trees and ground vegetation on both sides of the riverbank, dominated by Salicaceae and Betulaceae trees.
SP5	Pogragjë	42°38′1″ N 20°37′26″ E	The river receives discharges from village houses, agriculture, and other sources.	Deciduous trees and ground vegetation on both sides of the riverbank dominated by Salicaceae and Betulaceae.
SP6	Klinë–Center	42°61′52.56″ N 20°57′57.35″ E	Inside the city of Klina, concrete river channel on both sides. Municipal wastewaters and agricultural, industrial runoff discharge in the river.	The riverbank is concreted on both sides.
SP7	Klinë–lower part of the city	42°35′48″ N 20°34′0″ E	Lower part of the city of Klina. Municipal wastewaters and agricultural, industrial runoff discharge in the river.	The riverbank is dominated by Salicaceae trees.
SP8	Klinë	42°36′41″ N20°34′05″ E	Municipal wastewaters and agricultural, industrial runoff discharge in the river.	The riverbank is dominated by Salicaceae and Betulaceae.

**Table 2 plants-11-01469-t002:** The classification of water quality based on Macrophyte Index for Rivers for sandy type of a river [16,53,59].

No.	Ecological Status Class	Lowland River Sandy and Organic	Quality Class of Water
1	Very good	≥44.5	I
2	Good	44.5–35.0>	II
3	Moderate	35.0–25.4>	III
4	Poor	25.4–15.8>	IV
5	Bad	≤15.8	V

**Table 3 plants-11-01469-t003:** Biological Status Boundary Values [16,56].

Status	EQR Values
High	0.8
Good	0.6
Moderate	0.4
Poor	0.2
Bad	<0.2

The colors present the ecological status of water bodies: Blue-High, Green-Good, Yellow- Moderate, Orange-Poor, Red-Bad.

**Table 4 plants-11-01469-t004:** Summary statistics of environmental variables for the eight sampling sites, showing the mean and the standard deviation for each physicochemical parameters and their ecological status according to standard (GD161).

Variable	Units	SP1	SP2	SP3	SP4	SP5	SP6	SP7	SP8	Minimum and Maximum		M ± SD		Variance
Water temperature	°C	12.1	15.6	17.0	19.1	18.5	17.1	18.1	12.7	12.1–19.1		16.2 ± 2.6		6.8
Air temperature		16.8	17.4	17.9	20.9	21.3	23	21.1	21.1	16.8–23		19.9 ± 2.24		5.04
Turbidity	NTU	2.8	23.5	28.4	30.6	26.9	25.5	18.6	6.8	2.8–30.6		20.3 ± 10.3		106.2
Electrical conductivity Ec	µS/cm	459	856	829	766	797	762	753	376	376–856		699.7 ± 179.0		32,064
Total dissolved solids (TDS)	mg/L	229	428	415	333	398	377	373	186	186–428		342.3 ± 88.8		7890.2
Total suspended solids (TSS)	mg/L	14	46	85	39	28	3.9	3.6	1.1	1.1–85		27.5 ± 28.7		826.9
pH	0–14	7.93	7.46	7.60	8.26	7.70	7.08	7.62	8.11	7.08–8.26		7.72 ± 0.37		0.141
Dissolved oxygen (DO)	mg/L	5.33	3.15	0.03	5.97	4.73	7.2	4.5	10.9	0.03–10.9		5.22 ± 3.13		9.8
Quality class according to GD161 standard		III	V	V	III	IV	II	IV	I			III		
Dissolved oxygen saturation (DOS)	%	70.5	44.9	0.3	89.4	70	82	53	117	0.3–117		65.8 ± 34.5		1195.9
Biochemical oxygen demand (BOD)	mg/L	1.5	22.6	42.2	33.5	25.8	9.0	9.5	3.9	1.5–42.2		18.5 ± 14.7		218.6
Quality class according to GD161 standard		I	V	V	V	V	III	III	III			III		
Chemical oxygen demand (COD)	mg/L	4.2	49.6	87.1	63.5	58.0	18	19	6.8	4.2–87.1		38.2 ± 30.4		924.5
Quality class according to GD161 standard		I	III	V	V	V	III	III	II			III		
Total organic carbon (TOC)	mg/L	1.1	17.0	28.8	19.7	18.3	6.3	6.5	1.9	1.1–28.8		12.4 ± 9.9		98.3
Nitrates (NO3^−^)	mg/L	2.5	8.1	0.7	26.2	17.2	6.9	5.6	4.8	0.7–26.2		9.0 ± 8.5		72.6
Quality class according to GD161 standard		I	III	I	V	V	III	III	II			III		
n,n-diethyltryptamine (DET)	mg/L	<0.1	<0.1	0.8	0.2	0.1	0.2	0.4	<0.1					
Orthophosphates (PO43^−^)	mg/L	0.01	0.04	0.32	0.18	0.14	0.10	0.18	0.06	0.01–0.32		0.13 ± 0.09		0.010
Quality class according to GD161 standard		I	I	V	V	V	V	V	I			V		
Total phosphorus (PT)	mg/L	0.04	0.64	1.28	0.99	0.77	0.28	0.32	0.13	0.04–1.28		0.56 ± 0.43		0.193
Quality class according to GD161 standard		I	II	V	III	III	I	I	I			II		
Ammonia (NH_4_^+^)	mg/L	0.03	0.32	1.65	0.88	0.77	0.84	1.02	0.041	0.03–1.65		0.69 ± 0.54		0.299
Quality class according to GD161 standard		I	I	III	II	I	II	II	I			I		
Nitrites (NO_2_^−^)	mg/L	0.02	0.22	1.10	0.84	0.63	1.14	3.75	0.26	0.02–3.75		0.99 ± 1.18		1.4
Quality class according to GD161 standard		I	II	V	V	V	V	V	V			V		
Sulphate (SO_4_^2−^)	mg/L	19.5	28.9	16.5	38.3	18.3	15.9	14.8	7.9	7.9–38.3		20.01 ± 9.4		88.5
Quality class according to GD161 standard		I	I	I	I	I	I	I	I			I		
Calcium (Ca^2+^)	mg/L	70.87	130.1	134.1	112.1	121.3	123.3	127.3	70.8	70.87–134.1		111.2 ± 25.7		663.9
Quality class according to GD161 standard		I	II	II	II	II	II	II	I			II		
Magnesium (Mg^2+^)	mg/L	15.1	26.7	18.7	27.2	20.7	22.4	28.2	15.1	15.1–28.2		21.76 ± 5.2		27.8
Quality class according to GD161 standard		I	I	I	I	I	I	I	I			I		
Sodium (Na^+^)	mg/L	6.55	8.20	10.5	7.58	8.01	8.64	8.89	6.55	6.55–10.5		8.12 ± 1.30		1.7
Quality class according to GD161 standard		I	I	I	I	I	I	I	I			I		
Potassium (K^+^)	mg/L	2.21	2.66	2.68	3.00	3.22	3.11	3.51	2.21	2.21–3.51		2.83 ± 0.46		0.220
Chloride (Cl^−^)	mg/L	7.1	35.5	65.3	54.6	46.1	19.8	23.4	4.26	4.26–65.3		32.0 ± 22.1		492.3

The colors present the ecological status of water bodies: Blue-High, Green-Good, Yellow-Moderate, Orange-Poor, Red-Bad.

**Table 5 plants-11-01469-t005:** List of plants and index values.

Taxon	Family	SP1	SP2	SP3	SP4	SP5	SP6	SP7	SP8
*Adoxa moschatellina* L.	Adoxaceae	+							
*Agrostis stolonifera* L.	Poaceae							+	
*Amblystegium riparium* (Hedw.) Schimp.	Amblystegiaceae	+							
*Berula erecta* Huds.	Apiaceae					+	+		
*Bidens tripartita* Bigelow, 1824	Asteraceae				+	+			
*Callitriche cophocarpa* Sendtn.	Plantaginaceae					+			
*Callitriche stagnalis* Scop.	Plantaginaceae						+		
*Cardamine amara* L.	Brassicaceae	+							
*Cardamine flexuosa* With.	Brassicaceae	+							
*Ceratophyllum demersum* L.	Ceratophyllaceae					+			
*Cinclidotus aquaticus* (Hedw.) B. & S.	Pottiaceae	+							
*Cinclidotus aquaticus* Bruch & W.P.Schimper, 1842	Cinclidotaceae				+				
*Cinclidotus fontinaloides* Palisot de Beauvois	Cinclidotaceae	+							
*Cratoneuron filicinum* Spruce	Amblystegiaceae				+				
*Epilobium hirsutum* L.	Onagraceae	+			+	+			
*Epilobium sp*.		+							
*Epilobium tetragonum* Lour.	Onagraceae				+	+			
*Galium palustre* L.	Rubiaceae	+							
*Glyceria maxima* (Hartm.) Holmb	Poaceae		+	+	+			+	
*Helosciadium repens* (Jacq.) Koch	Apiaceae						+		
*Juncus effusus* L.	Juncaceae						+	+	+
*Juncus inflexus* L.	Juncaceae			+					
*Leersia oryzoides* Michx.	Poaceae					+			
*Lemna minor* L.	Araceae		+	+		+	+	+	+
*Lycopus europaeus* L.	Lamiaceae				+	+			
*Lysimachia nummularia* L.	Primulaceae	+							
*Lysimachia vulgaris* L.	Primulaceae				+				+
*Lythrum salicaria* L.	Lythraceae	+							
*Marchantia polymorpha L. var aquatic*	Marchantiaceae	+							
*Mentha aquatica* L.	Lamiaceae				+	+	+	+	+
*Mentha longifolia* (L.) Huds.	Lamiaceae	+							
*Myriophyllum spicatum* L.	Haloragaceae						+		
*Nasturtium microphyllum* (Boenn. ex Rchb.) Rchb.	Brassicaceae	+							
*Nasturtium officinale* R.Br.	Brassicaceae	+					+		+
*Persicaria amphibia* (L.) Delarbre	Polygonaceae	+							
*Petasites japonicus* subsp. *Japonicus*	Asteraceae	+							
*Phalaris arundinacea* L.	Poaceae						+		+
*Plantago media* L.	Plantaginaceae	+							
*Polygonum lapathifolium* L.	Polygonaceae	+	+		+				
*Polygonum latifolium* Giesecke	Polygonaceae								
*Polygonum mite* Schrank	Polygonaceae	+	+	+					
*Persicaria glabra* (Willd.) M.Gómez	Polygonaceae		+						
*Potamogeton crispus* L.	Potamogetonaceae					+	+	+	+
*Potamogeton fluitans* Sm.	Potamogetonaceae						+		
*Potamogeton gramineus* L.	Potamogetonaceae						+		
*Potamogeton natans* Sturm	Potamogetonaceae		+		+	+			
*Potamogeton nodosus* Poir.	Potamogetonaceae							+	
*Ranunculus repens* S.Watson	Ranunculaceae	+			+	+			+
*Ranunculus aquatilis* (Dumort.) Bab.	Ranunculaceaea							+	
*Rorippa palustris* (L.) Besser	Brassicaceae	+			+	+			
*Rorippa sylvestris* (L.) Besser	Brassicaceae				+	+			
*Rumex aquaticus* Campd.	Polygonaceae		+	+					
*Rumex hydrolapathum* Campd.	Polygonaceae				+	+	+		
*Rumex longifolius* DC.	Polygonaceae					+			
*Rumex palustris* Sm.	Polygonaceae			+					
*Schoenoplectus lacustris* (L.) Palla	Cyperaceae		+				+		
*Scrophularia auriculata* L.	Scrophulariaceae	+							+
*Scrophularia nodosa* L.	Scrophulariaceae					+			
*Scrophularia umbrosa* Salzm. ex Benth	Scrophulariaceae	+							
*Sparaganium erectum* L	Typhaceae		+		+	+	+		
*Stratiotes aloides* L.	Hydrocharitaceae							+	
*Stuckenia pectinata* (L.) Börner	Potamogetonaceae							+	
*Trichophorum cespitosum* (L.) Schur, 1853	Cyperaceae					+	+		
*Typha angustifolia* Eckl. & Zeyh. ex Rohrb., 1869	Typhaceae	+	+	+	+	+			
*Typha latifolia* G.Mey.	Typhaceae						+	+	+
*Veronica anagallis-aquatica* L.	Plantaginaceae	+		+			+		
*Veronica beccabunga* L.	Plantaginaceae					+	+		+
MIR (Macrophyte Index for Rivers)		45	44	25	40	76	57	20	23
Quality class of water		I	II	IV	II	I	I	IV	IV
RMNI (River Macrophyte Nutrient Index)		3.44	5.95	8.56	4.77	5.81	5.62	7.49	5.85
The ecological quality ratio for the parameter RMNI		1	0.61	0.21	0.80	0.63	0.68	0.38	0.63
RMHI (River Macrophyte Hydraulic Index)		3.57	6.27	8.42	5.96	5.77	5.61	8.5	5.20
The ecological quality ratio for the parameter RMHI		1	0.57	0.24	0.92	0.65	0.68	0.23	0.74

+ present.

**Table 6 plants-11-01469-t006:** The species density in each sampling point based on Kohler’s 5-point scale.

	SP1	SP2	SP3	SP4	SP5	SP6	SP7	SP8
Water depth	20–40 cm	60–100 cm	50–100 cm	40–60 cm	40–60 cm	60 cm	1.5 m	25 cm
Submersed plant density (0–5)	1	4	2	3	4	5	3	1
Plant density (0–5)	1	3	1	3	4	5	3	1
Cover (0–5)	1	4	2	2	4	4	2	1
Substrate	Gravel and rock	Silt and clay	Gravel and rock	Gravel and rock	Gravel and rock	Silt and clay	Silt and clay	Gravel and rock
Detritus	Present	Present	Present	Present	Present	Present	Present	Absent
Habitate	Terrestrial	Aquatic	Terrestrial	Terrestrial	Aquatic	Aquatic	Aquatic	Aquatic
S		+		+	+	+	+	+
F		+		+	+	+	+	+
E	+	+	+	+	+	+	+	+

E—Emergent, F—Floating, S—Submerged.

**Table 7 plants-11-01469-t007:** Pearson’s correlation between macrophyte indices and physicochemical parameters of water.

	Plant Density	RMNI	RMHI	EQR-RMNI	EQR-RMHI	MIR
Water temperature	0.714 *	0.406	0.606	−0.400	−0.389	0.241
Turbidity	0.726 *	0.436	0.555	−0.431	−0.358	0.286
Electrical conductivity Ec	0.780 *	0.483	0.626	−0.482	−0.516	0.272
Total dissolved solids (TDS)	0.765 *	0.545	0.639	−0.545	−0.600	0.281
Total suspended solids (TSS)	0.003	0.459	0.446	−0.469	−0.349	−0.082
pH	−0.652	−0.355	−0.279	0.344	0.470	−0.288
Dissolved oxygen (DO)	−0.202	−0.496	−0.569	0.505	0.554	0.009
Dissolved oxygen saturation (DOS)	−0.112	−0.640	−0.649	0.646	0.692	0.160
Biochemical oxygen demand (BOD)	0.245	0.474	0.534	−0.481	−0.336	0.054
Chemical oxygen demand (COD)	0.268	0.484	0.527	−0.492	−0.358	0.106
Total organic carbon (CTO)	0.283	0.515	0.552	−0.523	−0.396	0.091
Nitrates (NO_3_^−^)	0.424	−0.332	−0.117	0.332	0.408	0.430
Orthophosphates (PO_4_^3−^)	0.098	0.762 *	0.805 *	−0.760 *	−0.649	−0.268
Total phosphorous (PT)	0.238	0.503	0.562	−0.509	−0.364	0.031
Ammonia (NH_4_^+^)	0.330	0.746 *	0.812 *	−0.740 *	−0.686	−0.127
Nitrites (NO_2_^−^)	0.222	0.557	0.730 *	−0.548	−0.678	−0.398
Suphates (SO_4_^2−^)	0.340	−0.338	−0.064	0.335	0.350	0.211
Calcium (Ca^2+^)	0.741 *	0.652	0.760 *	−0.649	−0.688	0.135
Magnezium (Mg^2+^)	0.670	0.223	0.518	−0.217	−0.317	−0.015
Sodium (Na^+^)	0.368	0.847 **	0.850 **	−0.842 **	−0.834 *	−0.160
Potassium (K^+^)	0.745 *	0.374	0.574	−0.365	−0.465	0.227
Chloride (Cl^−^)	0.333	0.472	0.564	−0.478	−0.362	0.114

** Correlation is significant at the 0.01 level (2-tailed), * correlation is significant at the 0.05 level (2-tailed).

## Data Availability

The authors confirm that all data underlying the findings of this study are available within the article without any restriction.

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
