# Peer review of "The Macrophyte Indices for Rivers to Assess the Ecological Conditions in the Klina River in the Republic of Kosovo"

_plants, 2022, doi:10.3390/plants11111469_

Round 1
Reviewer 1 Report
In the manuscript “
The macrophyte indices for rivers to assess the ecologi-cal conditions in the Klina River in the Republic of Kosovo « authors Pajtim Bytyçi, Albona Shala-Abazi, Ferdije ZhushI-Etemi, Giuseppe Bonifazi, Mimoza Hyseni-Spahiu, Osman Fetoshi, Hazir Çadraku, Fidan Feka, Fadil Millaku, evaluated, how aquatic nutrients affect macrophyte species diversity, cover, and density in the Klina River.
Abstract
OK
settlements and various industrial activities.Plant… space is missing
the following sentence is not clear:
Plant density has been found to be significantly correlated with temperature and other parameters: Turbidity, total dissolved solids and potassium (p <0.05) and a strong positive correlation (p < 0.01) of RMNI, RMHI, EQR RMNI with Na+ and EQR RMHI (p < 0.01) with Na+ also positive correlation (p < 0.05) RMNI and RMHI indices with orthophosphates and ammonia.
Key words Are OK.
Introduction
Is too long.
Reference are missing in the first paragraph.
be expected.So… space is missing
Materials and Methods
(Kohler,1978): 1=very rare,2=rare,3=common,4=frequent, 5=abundant,predominant. Spaces are missing
Explain the MIR, RMNI, RMHI, EQR RMNI, EQR RMHI!
WHAT IS GD161 standard?
Results and Discussion
First part is only results, not discussion!!
Electrical conductivity (EC) The lowest value… The should be written in small letter
Lemna minor should be written in italic
Wang et al full stop is missing
Specific comments
In the Table 1, full stop is missing in certain places.
The idea of MS is interesting. However, the indices are not explained, and results and discussion section should be rewritten.
My suggestions: major revision
Author Response
Dear reviewers,
First of all, we want to thank you for the comments and suggestion given to our Manuscript.
We tried to answer to all your comments, and hope our answer will be appreciated and contribute for the acceptance of our Manuscript for publication.
Answers to the Reviewer 1.
*settlements and various industrial activities.Plant… space is missing
Answer
Corrected
the following sentence is not clear:
Plant density has been found to be significantly correlated with temperature and other parameters: Turbidity, total dissolved solids and potassium (p <0.05) and a strong positive correlation (p < 0.01) of RMNI, RMHI, EQR RMNI with Na+ and EQR RMHI (p < 0.01) with Na+ also positive correlation (p < 0.05) RMNI and RMHI indices with orthophosphates and ammonia.
Answer
Our conclusion in the Abstract is based on the results about correlation between physic-chemical parameters with Macrophyte indices and Ecological Quality ratio.
Please ,see the text below.
However, we can see a significant correlation of these variables in the table with water temperature, turbidity, electrical conductivity, total dissolved solids, orthophosphates, ammonia, nitrites, calcium, sodium and potassium. There is a strong positive correlation (p < 0.01) of Natrium (Na+) with RMNI, RMHI and EQR RMNI and EQRRMHI (p < 0.05), which means if natrium increase, these parameters will increase too. A positive correlation (p < 0.05), is shown between plant density and water temperature, NTU, TDS and K. There is a strong positive correlation (p < 0.01) of RMNI, RMHI, EQR RMNI with Na+ and EQR RMHI (p < 0.01), with Na+ also positive correlation (p < 0.05) RMNI and RMHI indices with Orthophosphates, and Ammonia. EQR RMNI are in negative correlation (significance p < 0.01) with Na+, indices EQRRMNI, EQRRMHI, Orthophosphates and Na+ (p < 0.05).
Introduction
Is too long.
Answer
The Introduction is shorten , as it is required by the reviewer.
Reference are missing in the first paragraph.
Answer: First paragraph is deleted.
be expected.So… space is missing
Answer: corrected
Materials and Methods
(Kohler,1978): 1=very rare,2=rare,3=common,4=frequent, 5=abundant,predominant. Spaces are missing
Answer: corrected
Explain the MIR, RMNI, RMHI, EQR RMNI, EQR RMHI!
Answer: Below we described the calculated indices and the formulas used for their calculation.
Macrophyte Index for Rivers (MIR) is calculated with the Equation (1):
MIR [1]
where: MIR – value of the Macrophyte Index for Rivers at the sampling site, N – number of
species at the sampling site; Li – indicator value for the i-th taxon , Wi – weighting factor for
the i-th taxon; Pi – ratio of coverage for the i-th taxon [51, 52, 55].
Table 2. The classification of water quality based on Macrophyte Index for Rivers for sandy type of a river [52]
No. |
Ecological status class |
Lowland river sandy and organic |
Quality class of water |
1 |
Very good |
≥ 44.5 |
I |
2 |
Good |
44.5 – 35.0> |
II |
3 |
Moderate |
35.0 – 25.4> |
III |
4 |
Poor |
25.4 – 15.8> |
IV |
5 |
Bad |
≤ 15.8 |
V |
River Macrophytes Nutrient Index (RMNI)
This index measures the aquatic plants (macrophytes) growth in the river in relation to nutrients. Depending on the amount of nutrients in the water, it value ranges from 1-10, [56, 33]. RMNI is calculated with the formula:
RMNI [2]
River Macrophyte Hydraulic Index (RMHI)
This index measures the macrophytes grow related to the river flow rate, and is expressed on the scale from 1-10, the higher the flow rate the higher the energy and vice versa [56, 33]. RMHI is calculated according to the formula:
RMHI [3]
How do we decide the Biological status?
Ecological Quality Ratio (EQR), is axpresed as a numeric value that ranges from 1 (natural or near natural state) to 0 (highly degraded by pollution or other disturbance) [56, 33].
This is subdivided equally into the five quality categories, as required by the WFD.
EQR for the parameter RMNI, should be calculated using the following equation:
EQRRMNI = (observed value of RMNI – worst possible RMNI) ÷ (reference value for RMNI – worst possible RMNI) [4]
EQR for the parameter RMHI, should be calculated using the following equation:
EQRRMHI = (observed value of RMHI - worst possible RMHI) ÷ (Reference value of RMHI - worst possible RMHI). [5]
Table 3. Biological Status Boundary Values [56]
Status |
EQR Values |
High |
0.8 |
Good |
0.6 |
Moderate |
0.4 |
Poor |
0.2 |
Bad |
<0.2 |
WHAT IS GD161 standard?
This standard is part of the ANNEX I of EWFD (Quality elements and physico-chemical quality standards for assessment of ecological status of surface water in Romania, 2006) (GD 161)
Electrical conductivity (EC) The lowest value… The should be written in small letter
Answer: corrected
Lemna minor should be written in italic
Answer: corrected
Wang et al full stop is missing
Answer: corrected
In the Table 1, full stop is missing in certain places.
Answer: corrected
The idea of MS is interesting. However, the indices are not explained, and results and discussion section should be rewritten.
Answer: The indices are explained, in some parts, the results and the discussion are rewritten

Reviewer 2 Report
Major comments:
- Evaluation of macrophytes diversity and density gives a wide characteristic of water ecology but such a work needs much time. Taking into account that Lemna minor contrary to other plants is present almost in all sampling places (6 out of 8) one may suggest that utilization of only this species may become optimal for water quality evaluation- is it possible to widen the discussion in this respect?
- Material and Methods section should contain more information about the parameters used: add to Material and methods section the descriptions of MIR, RMNI, RMHI, EQR RMNI and EQR RMHI calculation. The text of the Discussion section should also contain a brief typical characteristics of these parameters, their benefits.
- While discussing Table 5 data it is desirable not to repeat in the text concrete values indicated in the Table
Minor comments:
1) Abstract: decipher ‘EQR'
2) Introduction: delete repetition: ‘The assessment and classification of freshwater ecosystems through the assessment of their ecological status was introduced as a monitoring task almost 20 years ago by the Water Framework Directive (WFD)’ and further: ‘In order to monitor and assess rivers, many scientific studies have recently been carried out in different parts of the world for biological monitoring and aquatic ecosystem study based on macrophytes as one of the biological elements for the assessment of the ecological status of rivers required by the EU Water Framework Directive (WFD).
3) Page 4: ‘EWFD requirements’- or ‘WFD requirements’?
4) Usually Material and Methods section in Plants Journal is placed after discussion at the end of the manuscript
5) as ‘the Klina river is the second biggest tributary of the Drini i Bardhë river basin’- are there any data on water quality of the Drini i Bardhë river”?
6) Table 1 a)column ‘Hydro morphology’: ‘Macrophyte present’ may be deleted, as they are present in all samples sites- just indicate the fact in the text b) add the title of the first column
7) Page 1 line 30: please decipher ‘MIR, RMNI, RMHI, EQR RMNI, EQR RMHI’
8) Line 44 and further throughout the text change ‘natrium’ to ‘sodium’
9) Lines 36-46- should be placed before statistics
10)Table 2- decipher abbreviation (MIR etc) in the Table notes
11) add the data about MIR, RMNI, RMHI, EQR RMNI and EQR RMHI peculiarities and benefits, indicate why these indexes were chosen
12) page 4 Line 106: decipher ‘DO, BOD, COD, TOC, Nitrates and MTS’
13) Table 3- decipher ‘MIR, RMNI, RMHI, EQR RMNI and EQR RMHI’ under the Table
14) Line 115 change ‘natrium’ to ‘sodium’
15) Fig.2- indicate the axis units
16) Table 5 a) misprints: ‘Chemical Ooxygen Demand’; ‘Magnezium'
b)delete abbreviations in column 1 (TSS, TDS etc)
c)column 3- add the title (units)
d)please simplify the Table data: combine ‘minimum’ and ‘maximum’ columns to one: ‘ parameter range’ and ‘mean’ and ‘standard deviation’ to ‘M±SD’
- e) for each parameter number of decimal places should be the same (for instance mean water temperature should be not ‘16.275’ but ’16.3’. TDS should be 342±89’ but not 3/ 88.8/ Pay attention to pH especially: M±SD should be ‘7.72±0.38’, but not ‘7.72±.37693’. The same with K, Cl, Ca
- f) unify numbers- for instance, ‘orthophosphates’ line- ‘0.066/ .011/ .320/ .13/.09…’ change to ‘0.0066/0.011-0.320/0.13±09
17) page 10 line 10 ‘Excess of nutrients N and P, in water body cause overproduction’ change to ‘Excess of nutrients N and P, in water body caused overproduction’
18) Page 10, line 32- TOC- is indicated in Table 5, not Table 4
19) Page 10 lines 36-37 ‘. It has been proven that Macrophytes affect the reduction of Nitrate in water while, depending on their ability to produce carbon for denitrification’- Is it possible to correlate the results of nitrates/nitrites water concentrations with macrophytes influence?
20) Page 10 line 52 – please check ‘nitrates’ or ‘nitrites’? It seems to be a misprint. Or you are speaking about ammonia????
21) Page 10 line 51 change ‘Wang et al (2021)’ to ‘Wang et al [77]’
22) Fig.3- add units for the ‘distance’ parameter
23) Page 12 lines 87-92- repetition of the previous 82-86 lines
24) Author contribution and funding, data availability statement, acknowledgement, conflict of interests information should be added
25) Reference list:
- i) please, check the whole list according to the journal guidelines: use Italics for the journals titles and volumes and bold letters for the year;
- ii) please, delete repetitions- refs 38, 40;
iii) use abbreviation of the journal’s title : refs.16, 30, 71,76 etc
Author Response
Dear reviewers,
First of all, we want to thank you for the comments and suggestions given to our Manuscript.
Answers to the Reviewer 2
Abstract: decipher ‘EQR'
Answer: corrected
Introduction: delete repetition: ‘The assessment and classification of freshwater ecosystems through the assessment of their ecological status was introduced as a monitoring task almost 20 years ago by the Water Framework Directive (WFD)’ and further: ‘In order to monitor and assess rivers, many scientific studies have recently been carried out in different parts of the world for biological monitoring and aquatic ecosystem study based on macrophytes as one of the biological elements for the assessment of the ecological status of rivers required by the EU Water Framework Directive (WFD).
Answer: Paragraph deleted
Page 4: ‘EWFD requirements’- or ‘WFD requirements’?
Answer: corrected
4) Usually Material and Methods section in Plants Journal is placed after discussion at the end of the manuscript
Answer: We based our Manuscript structure in few articles published in Plant Journal.
as ‘the Klina river is the second biggest tributary of the Drini i Bardhë river basin’- are there any data on water quality of the Drini i Bardhë river”?
Answer: There are few publication on water quality of Drini i Bardhë river basin based on other biological quality elements, such as macroinvertebrates and fish, but not based on Macrophytes
Table 1 a)column ‘Hydro morphology’: ‘Macrophyte present’ may be deleted, as they are present in all samples sites- just indicate the fact in the text b) add the title of the first column
Answer: corrected
Page 1 line 30: please decipher ‘MIR, RMNI, RMHI, EQR RMNI, EQR RMHI’
Answer: we gave more details in Material and Methods
Line 44 and further throughout the text change ‘natrium’ to ‘sodium’
Answer: changed
Lines 36-46- should be placed before statistics
Answer: Thank you for the suggestion! We placed the suggested text before the statistics.
Table 2- decipher abbreviation (MIR etc) in the Table notes
Answer: corrected
add the data about MIR, RMNI, RMHI, EQR RMNI and EQR RMHI peculiarities and benefits, indicate why these indexes were chosen
Answer: These indexes MIR, RMNI are chosen since they indicate the presence of nutrients in the water bodies and thus stimulate the Macrophyte growth, whereas RMHI indicates the vegetation growth depending on water velocity and substrate stability.
page 4 Line 106: decipher ‘DO, BOD, COD, TOC, Nitrates and MTS’
Answer: corrected
Table 3- decipher ‘MIR, RMNI, RMHI, EQR RMNI and EQR RMHI’ under the Table
Answer: corrected
Line 115 change ‘natrium’ to ‘sodium’
Answer: corrected
Table 5 a) misprints: ‘Chemical Ooxygen Demand’; ‘Magnezium'
Answer: corrected
Table 5 a) misprints: ‘Chemical Ooxygen Demand’; ‘Magnezium'
b)delete abbreviations in column 1 (TSS, TDS etc)
Answer: corrected
c)column 3- add the title (units)
Answer: corrected
please simplify the Table data: combine ‘minimum’ and ‘maximum’ columns to one: ‘ parameter range’ and ‘mean’ and ‘standard deviation’ to ‘M±SD’
Answer: corrected
for each parameter number of decimal places should be the same (for instance mean water temperature should be not ‘16.275’ but ’16.3’. TDS should be 342±89’ but not 3/ 88.8/ Pay attention to pH especially: M±SD should be ‘7.72±0.38’, but not ‘7.72±.37693’. The same with K, Cl, Ca
Answer: corrected
unify numbers- for instance, ‘orthophosphates’ line- ‘0.066/ .011/ .320/ .13/.09…’ change to ‘0.0066/0.011-0.320/0.13±09
Answer: corrected
page 10 line 10 ‘Excess of nutrients N and P, in water body cause overproduction’ change to ‘Excess of nutrients N and P, in water body caused overproduction’
Answer: corrected
Page 10, line 32- TOC- is indicated in Table 5, not Table 4
Answer: corrected
Page 10 lines 36-37 ‘. It has been proven that Macrophytes affect the reduction of Nitrate in water while, depending on their ability to produce carbon for denitrification’- Is it possible to correlate the results of nitrates/nitrites water concentrations with macrophytes influence?
Answer: In the lines 36-37 it is stated that the ability of macrophytes present species to efficiently remove the nitrites depends on their ability to produce carbon for denitrification. In this case the correlation nitrate/nitrites concentration with macrophyte species that produce carbon is negative.
Page 10 line 52 – please check ‘nitrates’ or ‘nitrites’? It seems to be a misprint. Or you are speaking about ammonia????
Answer: Nitrites (NO2-)
Page 10 line 51 change ‘Wang et al (2021)’ to ‘Wang et al [77]’
Answer: corrected
Page 12 lines 87-92- repetition of the previous 82-86 lines
Answer: deleted
24) Author contribution and funding, data availability statement, acknowledgement, conflict of interests information should be added
Answer:
all of these have been added
25) Reference list:
- i) please, check the whole list according to the journal guidelines: use Italics for the journals titles and volumes and bold letters for the year;
- ii) please, delete repetitions- refs 38, 40;
iii) use abbreviation of the journal’s title : refs.16, 30, 71,76 etc
Answer: corrected

Round 2
Reviewer 1 Report
In the manuscript “ The macrophyte indices for rivers to assess the ecological conditions in the Klina River in the Republic of Kosovo « authors Pajtim Bytyçi, Albona Shala-Abazi, Ferdije ZhushI-Etemi, Giuseppe Bonifazi, Mimoza Hyseni-Spahiu, Osman Fetoshi, Hazir Çadraku, Fidan Feka, Fadil Millaku, evaluated, how aquatic nutrients affect macrophyte species diversity, cover, and density in the Klina River.
Abstract
OK
the following sentence is STILL not clear. May you have only put the punctations on the right places!
Plant density has been found to be significantly correlated with temperature and other parameters: Turbidity, total dissolved solids and potassium (p <0.05) and a strong positive correlation (p < 0.01) of RMNI, RMHI, EQR RMNI with Na+ and EQR RMHI (p < 0.01) with Na+ also positive correlation (p < 0.05) RMNI and RMHI indices with orthophosphates and ammonia.
Key words Are OK.
Introduction OK
Materials and Methods OK
Results and Discussion
In the followings statement, full stop and space is missing: The variation of orthophosphates ranged from 0.011 in SP1 to 0.320 in SP3Orthophosphate levels place the Klina River in the fifth class and in poor water quality.
In the followings paragraph, English is not clear: Of the 67 macrophyte species present in the Klina River, only in SP1 were prevalent:…
Write please, what + Means in the Table 5
In the Table 6 - Plant density is meant to the whole SP? Difference between submersed plant density, and plant density in Table 6?? Difference between plant density and abundance? Where is abundance of the species?
In the followings paragraph, English is not clear There is a strong positive correlation (p < 0.01) of Sodium (Na+) with RMNI, RMHI and EQR RMNI and EQRRMHI (p < 0.05), which means if Sodium increase, these parameters will increase too. A positive correlation (p < 0.05), is shown between plant density and water temperature, NTU, TDS and K. There is a strong positive correlation (p < 0.01) of RMNI, RMHI, EQR RMNI with Na+ and EQR RMHI (p < 0.01), with Na+ also positive correlation (p < 0.05) RMNI and RMHI indices with Orthophosphates, and Ammonia. EQR RMNI are in negative correlation (significance p < 0.01) with Na+, indices EQRRMNI, EQRRMHI, Orthophosphates and Na+ (p < 0.05).
There should be more discussion with the literature data!
Specific comments
Do you estimate ecological conditions in the Klina river as it is in the title, or the quality of water as it is in the Results and Discussion section? Carefully read your MS, again and put commas and spaces on the right positions, and capital and small letters,…!
The idea of MS is interesting. However, results and discussion section should be deeper.
My suggestions: minor revision
Author Response
Dear editor and reviewers,
First of all, we want to thank you for the comments and suggestions given to our Manuscript.
We tried to answer all your comments, and hope our answer will be appreciated and contribute to the acceptance of our Manuscript for publication.
Best regards,
Prof. Dr sc. Albona Shala-Abazi
the following sentence is STILL not clear. May you have only put the punctations on the right places!
Plant density has been found to be significantly correlated with temperature and other parameters: Turbidity, total dissolved solids and potassium (p <0.05) and a strong positive correlation (p < 0.01) of RMNI, RMHI, EQR RMNI with Na+ and EQR RMHI (p < 0.01) with Na+ also positive correlation (p < 0.05) RMNI and RMHI indices with orthophosphates and ammonia.
Answer:
There is a positive correlation (p < 0.05) between Water Temperature, Turbidity, Electrical conductivity (EC), Total dissolved solids (TDS), Orthophosphates (PO43-), Ammonia (NH4+), Nitrites (NO2-), Calcium (Ca2+) and Potassium (K+), with Plant density, RMNI, RMHI, EQR-RMNI, EQR-RMHI and MIR. Sodium (Na+) has a stronger positive correlation (p<0.01) with RMNI and RMHI indices and a negative correlation with EQR-RMNI and EQR-RMHI.
Results and Discussion
*In the followings statement, full stop and space is missing: The variation of orthophosphates ranged from 0.011 in SP1 to 0.320 in. SP3 Orthophosphate levels place the Klina River in the fifth class and in poor water quality.
Answer:
Corrected
*In the followings paragraph, English is not clear: Of the 67 macrophyte species present in the Klina River, only in SP1 were prevalent:…
Answer:
We clarified in the text which species were present only in one of the localities, and were absent
*Write please, what + Means in the Table 5
Answer:
+ means the species is present
*In the Table 6 - Plant density is meant to the whole SP? Difference between submersed plant density, and plant density in Table 6?? Difference between plant density and abundance? Where is abundance of the species?
Answer:
The density of Submerse plants is separately calculated and includes only the rooted macrophytes that grow under water, whereas the plant diversity means the diversity of all macrophyte species.
*In the followings paragraph, English is not clear There is a strong positive correlation (p < 0.01) of Sodium (Na+) with RMNI, RMHI and EQR RMNI and EQRRMHI (p < 0.05), which means if Sodium increase, these parameters will increase too. A positive correlation (p < 0.05), is shown between plant density and water temperature, NTU, TDS and K. There is a strong positive correlation (p < 0.01) of RMNI, RMHI, EQR RMNI with Na+ and EQR RMHI (p < 0.01), with Na+ also positive correlation (p < 0.05) RMNI and RMHI indices with Orthophosphates, and Ammonia. EQR RMNI are in negative correlation (significance p < 0.01) with Na+, indices EQRRMNI, EQRRMHI, Orthophosphates and Na+ (p < 0.05).
Answer:
Our results show that sodium (Na+) positively impacts the nutrient concentration in the water, however it has negative effect in ecological status based on RMNI and RMHI.
Specific comments
*Do you estimate ecological conditions in the Klina river as it is in the title, or the quality of water as it is in the Results and Discussion section? Carefully read your MS, again and put commas and spaces on the right positions, and capital and small letters,…!
The idea of MS is interesting. However, results and discussion section should be deeper.
Answer:
The aim of our MS was to estimate Ecological conditions in the Klina river using Macrophyte based indices. Since the calculated values of indices indicate a water quality category therefore we discussed both the water quality and the Ecological status.
